# Global burden of disease due to rifampicin-resistant tuberculosis: a mathematical modeling analysis

Nicolas A. Menzies [1,2] ✉, Brian W. Allwood[3,15], Anna S. Dean [4,15],
Pete J. Dodd [5,15], Rein M. G. J. Houben[6,7,15], Lyndon P. James[2,8,15],
Gwenan M. Knight [9,15], Jamilah Meghji[10,15], Linh N. Nguyen[4,15],
Andrea Rachow[11,12,13,15], Samuel G. Schumacher[4,15], Fuad Mirzayev [4] &
Ted Cohen [14]

In 2020, almost half a million individuals developed rifampicin-resistant tuberculosis (RR-TB). We estimated the global burden of RR-TB over the lifetime of affected individuals. We synthesized data on incidence, case detection, and treatment outcomes in 192 countries (99.99% of global tuberculosis). Using a mathematical model, we projected disability-adjusted life years (DALYs) over the lifetime for individuals developing tuberculosis in 2020 stratified by country, age, sex, HIV, and rifampicin resistance. Here we show that incident RR-TB in 2020 was responsible for an estimated 6.9 (95% uncertainty interval: 5.5, 8.5) million DALYs, 44% (31, 54) of which accrued among TB survivors. We estimated an average of 17 (14, 21) DALYs per person developing RR-TB, 34% (12, 56) greater than for rifampicin-susceptible tuberculosis. RR-TB burden per 100,000 was highest in former Soviet Union countries and southern African countries. While RR-TB causes substantial short-term morbidity and mortality, nearly half of the overall disease burden of RR-TB accrues among tuberculosis survivors. The substantial long-term health impacts among those surviving RR-TB disease suggest the need for improved post-treatment care and further justify increased health expenditures to prevent RR-TB transmission.

In 2021, it is estimated that 450 thousand individuals developed rifampicin-resistant tuberculosis (RR-TB)[1]. Rifampicin is one of the key drugs in the WHO-recommended first-line tuberculosis treatment regimen, and rifampicin resistance represents a major challenge for effective tuberculosis treatment. For individuals diagnosed with resistant disease, the most commonly used treatment regimens have required an extended duration of therapy, using drugs that are both more toxic and less effective compared to the first-line regimen (although recent changes in WHO policy may change this over time[2]). As a result, individuals treated for RR-TB experience lower cure rates and higher mortality rates compared to individuals with drug-susceptible tuberculosis treated with the first-line regimen[3]. Moreover, many individuals diagnosed with tuberculosis are not tested for drug resistance, such that many of those with a drug-resistant strain will receive an inappropriate treatment regimen, leading to high rates of treatment failure and recurrence[4,5]. Due to these factors, many individuals who develop RR-TB will die during the disease episode or experience a prolonged period of ill health before achieving a cure.

Accumulating evidence indicates that tuberculosis survivors frequently experience ongoing disability[6,7], elevated mortality rates[8], income losses[9], and persistent social and psychological sequalae[10]. These ongoing effects contribute a substantial fraction of the overall

disease burden caused by tuberculosis[11,12]. Evidence suggests that these post-TB sequelae are more common and severe for individuals surviving RR-TB[13,14]. Due to delays in initiating appropriate treatment[15] and the lower efficacy of treatment regimens, individuals treated for RR-TB are likely to accumulate greater lung damage before the disease is effectively controlled by treatment. Moreover, individuals with RR-TB may receive several treatment courses before achieving a cure due to receipt of inappropriate first-line treatment or lower success rates even when appropriate treatment is available, with additional lung damage occurring during this extended disease period. The greater toxicity of drugs included in RR-TB regimens can also cause long-term sequelae. A meta-analysis of data on tuberculosis survivors found greater lung impairment among individuals surviving multi-drug resistant (MDR) tuberculosis compared to individuals with drug-susceptible tuberculosis[14], and high levels of impairment among MDR-TB survivors have been reported across a range of functional domains[16].

While global RR-TB incidence is estimated to be declining[1], these trends—as well as the absolute fraction of incident tuberculosis cases with rifampicin resistance—vary substantially across world regions. Similarly, national tuberculosis programs vary greatly in their capacity to identify RR-TB and provide effective treatment. The development of better diagnostics and RR-TB regimens promises to improve patient care in the future, yet the adoption of these new tools has been uneven. These differences between settings will produce differences in the magnitude of disease burden attributable to RR-TB. In this study, we synthesized evidence on the incidence, detection, and outcomes of RR-TB treatment across 192 countries and used mathematical modeling to simulate the lifetime burden of disease due to incident RR-TB in 2020, including health losses incurred during the disease episode as well as among individuals surviving tuberculosis. Based on this analysis, we report the number of disability-adjusted life years (DALYs) attributable to incident RR-TB in 2020, stratified by country, age, sex, HIV status, and disease stage.

## Results

### Global DALYs attributable to RR-TB

Incident RR-TB occurring in 2020 was estimated to result in 6.9 (95% uncertainty interval: 5.5, 8.5) million DALYs over the lifetime of affected individuals, representing 5.4% (4.4, 6.3) of lifetime DALYs attributed to all incident tuberculosis in this year. The majority (85.0% (79.8, 89.3)) of RR-TB DALYs were estimated to result from premature mortality (5.9 (4.8, 7.1) million DALYs), as compared to reductions in quality of life (1.1 (0.6, 1.6) million DALYs). Just over half (55.8% (46.0, 68.5)) of all RR-TB DALYs were estimated to be incurred during the disease episode (3.8 (3.2, 4.4) million DALYs), with the remainder resulting from post-TB sequelae among tuberculosis survivors (3.1 (1.8, 4.4) million DALYs). Figure 1 summarizes each contribution to overall RR-TB DALYs per case. We estimated an average of 17.3 (13.8, 20.6) DALYs per person developing RR-TB, 34.2% (11.8, 55.7) greater than for individuals who developed RS-TB. Of these excess DALYs, the majority resulted from additional mortality during and after the TB episode. Table 1 reports estimates of the lifetime disease burden attributable to incident RS-TB and RR-TB occurring in 2020, globally and per case.

### Burden of RR-TB by region, country, age, sex, and HIV status

Table 2 reports results stratified by region. RR-TB burden was estimated to be the largest in the South-East Asia Region (37.2% (32.6, 41.9) of the global total), followed by the African Region (24.8% (20.1, 31.0) of the global total). Taken together, the group of thirty high MDR/RR-TB burden countries identified by the WHO accounted for 6.0 (4.6, 7.4) million RR-TB DALYs, 85.9% (82.5, 88.4) of the global total. Per population, RR-TB DALYs were highest in former Soviet Union countries, and southern African countries. DALYs per incident RR-TB case were generally higher in sub-Saharan Africa. Figure 2 shows estimates for RR-TB DALYs per 100,000 of population, and Fig. 3 shows estimates for RR-TB DALYs per incident RR-TB case for each country included in the analysis. Supplementary Table S2 reports RR-TB DALYs and uncertainty esti-

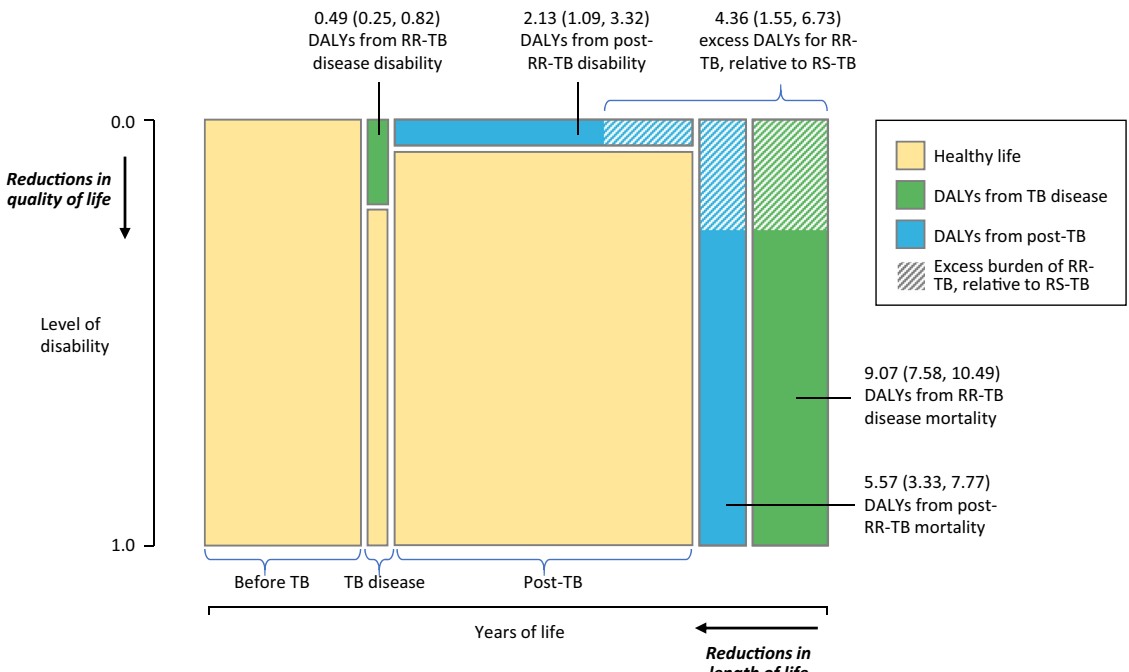

**Fig. 1 | Average DALYs per RR-TB case from increased disability and mortality rates attributable to tuberculosis, stratified by tuberculosis disease and post-tuberculosis period.** Area of each green and blue rectangle and shaded regions are proportional to the number of DALYs indicated, other dimensions not to scale.

Values in parentheses represent 95% uncertainty intervals. Total RR-TB DALYs per incident tuberculosis case are equal to the sum of these values. DALYs disability-adjusted life years, RR-TB rifampicin-resistant tuberculosis, RS-TB rifampicin-susceptible tuberculosis.

**Table 1 | Lifetime burden of tuberculosis stratified by RS-TB and RR-TB, globally and per incident tuberculosis case, 2020**

| Form of disease | Outcome | TB disease | Post-TB | Total |
|---|---|---|---|---|
| **Total global DALYs (millions)** | | | | |
| RS-TB | YLLs | 63.5 (55.5, 72.1) | 39.7 (25.6, 57.9) | 103.2 (86.4, 122.2) |
| | YLDs | 4.9 (2.5, 8.3) | 14.5 (8.3, 23.5) | 19.4 (12.4, 28.5) |
| | DALYs | 68.4 (59.7, 77.6) | 54.2 (35.6, 79.7) | 122.6 (100.8, 149.0) |
| RR-TB | YLLs | 3.6 (3.1, 4.3) | 2.2 (1.3, 3.2) | 5.9 (4.8, 7.1) |
| | YLDs | 0.2 (0.1, 0.3) | 0.9 (0.4, 1.4) | 1.1 (0.6, 1.6) |
| | DALYs | 3.8 (3.2, 4.4) | 3.1 (1.8, 4.4) | 6.9 (5.5, 8.5) |
| All disease | YLLs | 67.1 (58.8, 76.1) | 42.0 (27.6, 60.7) | 109.1 (91.9, 128.9) |
| | YLDs | 5.1 (2.6, 8.6) | 15.3 (8.8, 24.7) | 20.4 (13.1, 29.9) |
| | DALYs | 72.2 (63.2, 81.9) | 57.3 (37.7, 83.5) | 129.5 (107.1, 156.9) |
| **DALYs per incident tuberculosis case** | | | | |
| RS-TB | YLLs | 6.7 (5.9, 7.5) | 4.2 (2.8, 6.0) | 10.9 (9.3, 12.7) |
| | YLDs | 0.5 (0.3, 0.9) | 1.5 (0.9, 2.5) | 2.0 (1.3, 3.0) |
| | DALYs | 7.2 (6.4, 8.0) | 5.7 (3.7, 8.4) | 12.9 (10.8, 15.6) |
| RR-TB | YLLs | 9.1 (7.6, 10.5) | 5.6 (3.3, 7.8) | 14.6 (12.0, 17.1) |
| | YLDs | 0.5 (0.2, 0.8) | 2.1 (1.1, 3.3) | 2.6 (1.6, 3.9) |
| | DALYs | 9.6 (8.1, 11.0) | 7.7 (4.5, 10.7) | 17.3 (13.8, 20.6) |
| All disease | YLLs | 6.8 (6.0, 7.6) | 4.2 (2.8, 6.1) | 11.0 (9.4, 12.8) |
| | YLDs | 0.5 (0.3, 0.9) | 1.5 (0.9, 2.5) | 2.1 (1.3, 3.0) |
| | DALYs | 7.3 (6.5, 8.1) | 5.8 (3.8, 8.4) | 13.1 (11.0, 15.7) |

*YLLs* 'years of life lost' due to fatal disease consequences, *YLDs* 'years lived with disability' due to non-fatal disease consequences, *DALYs* disability-adjusted life years (the sum of YLLs and YLDs), *RR-TB* rifampicin-resistant tuberculosis, *RS-TB* rifampicin-susceptible tuberculosis.
Values in parentheses represent 95% uncertainty intervals.

**Table 2 | Global DALYs resulting from incident RR-TB in 2020 by country grouping**

| Region | RR-TB DALYs per 100,000 | DALYs per RR-TB case | Total RR-TB DALYs (millions) | Percent of global RR-TB burden (%) |
|---|---|---|---|---|
| African Region | 152.0 (111.4, 204.4) | 23.0 (18.4, 27.3) | 1.72 (1.26, 2.32) | 24.8 (20.1, 31.0) |
| Region of the Americas | 13.7 (9.0, 20.1) | 13.9 (10.6, 17.2) | 0.14 (0.09, 0.21) | 2.0 (1.4, 2.8) |
| Eastern Mediterranean Region | 83.6 (41.8, 138.2) | 14.8 (10.5, 19.4) | 0.63 (0.32, 1.04) | 9.0 (5.2, 14.2) |
| European Region, excluding former Soviet Union[a] | 3.2 (2.4, 4.1) | 11.7 (8.7, 14.7) | 0.02 (0.01, 0.03) | 0.3 (0.2, 0.4) |
| Former Soviet Union countries[a] | 290.7 (225.2, 369.1) | 14.2 (11.1, 17.4) | 0.87 (0.68, 1.11) | 12.6 (10.5, 15.0) |
| South-East Asia Region | 126.1 (99.9, 153.4) | 19.6 (15.9, 23.3) | 2.57 (2.04, 3.13) | 37.2 (32.6, 41.9) |
| Western Pacific Region | 50.5 (34.6, 71.1) | 12.2 (8.8, 15.8) | 0.97 (0.67, 1.37) | 14.0 (11.2, 17.5) |
| High MDR/RR-TB burden | 129.8 (101.0, 161.6) | 17.0 (13.3, 20.3) | 5.96 (4.63, 7.42) | 85.9 (82.5, 88.4) |
| Global | 88.8 (70.7, 109.2) | 17.3 (13.8, 20.6) | 6.93 (5.52, 8.53) | 100 (100, 100) |

*DALYs* disability-adjusted life years, *RR-TB* rifampicin-resistant tuberculosis.
Values in parentheses represent 95% uncertainty intervals.
[a]Former Soviet Union countries reported separately from the European Region, given their different epidemiology of tuberculosis drug resistance. High MDR/RR-TB countries include Angola, Azerbaijan, Bangladesh, Belarus, China, Democratic Republic of Congo, Indonesia, India, Kazakhstan, Kyrgyzstan, Moldova, Myanmar, Mongolia, Mozambique, Nigeria, Nepal, Pakistan, Peru, Philippines, Papua New Guinea, North Korea, Russia, Somalia, Tajikistan, Ukraine, Uzbekistan, Viet Nam, South Africa, Zambia, and Zimbabwe.

mates for each country, with India estimated to have the greatest absolute burden of RR-TB (1.7 (1.3, 2.1) million DALYs, 24.1% (20.3, 28.2) of global RR-TB DALYs), followed by Russia (0.5 (0.4, 0.6) million DALYs, 6.7% (5.4, 8.3) of global RR-TB DALYs). Lesotho was estimated to have the highest burden per population (773 (444, 1225) DALYs per 100,000), followed by South Africa (703 (449, 1030) per 100,000). Taken together, the 10 countries comprising continental southern Africa (Angola, Botswana, Eswatini, Lesotho, Namibia, Malawi, Mozambique, South Africa, Zambia, Zimbabwe) represented 11.0% (8.3, 14.5) of total global burden (0.76 (0.53, 1.08) million DALYs), with 416.3 (287.6, 590.5) DALYs per 100,000 and 22.2 (16.7, 27.7) DALYs per RR-TB case.

Table 3 reports RR-TB burden estimates stratified by age group, sex, and HIV status, with DALYs per incident case declining with age and with rates per 100,000 peaking in 35- to 44-year-olds. Men experienced higher disease burden compared to women (109.2 (85.8, 135.4) vs. 68.1 (52.7, 85.3) DALYs per 100,000) due to elevated TB incidence rates. For individuals with HIV, the burden of RR-TB was substantially higher than for HIV-uninfected populations, with RR-TB DALYs per 100,000 estimated to be 39.2 (30.2, 51.7) times higher than for HIV-uninfected populations and 15.9% (12.8, 20.0) of total RR-TB DALYs were incurred by individuals with HIV.

**Sensitivity of results to parameter uncertainty**
Supplementary Table S3 reports PRCCs describing the sensitivity of RR-TB DALY estimates to uncertainty in individual parameters. The five

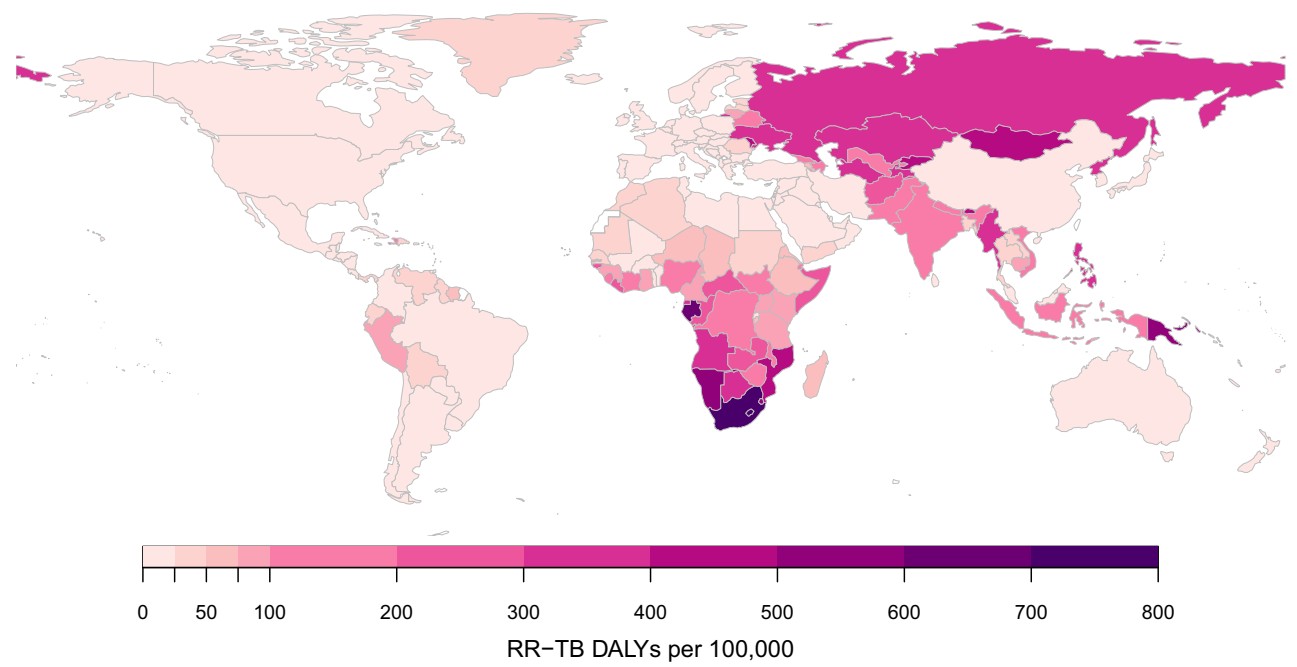

**Fig. 2 | Estimated DALYs per 100,000 due to incident RR-TB in 2020 by country.** DALYs disability-adjusted life years, RR-TB rifampicin-resistant tuberculosis.

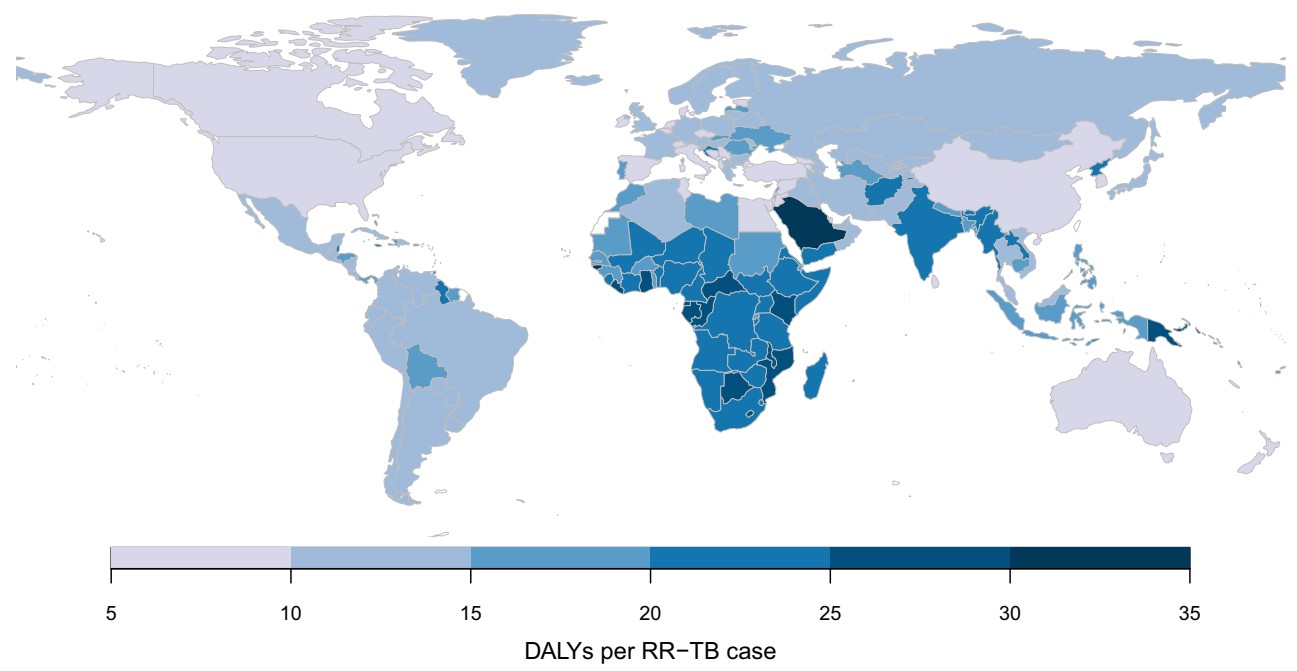

**Fig. 3 | Estimated DALYs per incident case due to incident RR-TB in 2020 by country.** DALYs disability-adjusted life years, RR-TB rifampicin-resistant tuberculosis.

most influential parameters included those determining the magnitude of mortality rate ratios and disability weights due to post-TB sequelae among individuals surviving tuberculosis disease, parameters determining the absolute number of RR-TB cases (total tuberculosis incidence, RR-TB prevalence in new tuberculosis cases), and the parameter determining case fatality for individuals with RR-TB inappropriately receiving a first-line regimen.

### Sensitivity of results to alternative analytic assumptions
Supplementary Table S4 reports global-level results for six alternative analytic specifications, as compared to the results of the main analysis.

The magnitude of DALY estimates was generally robust to these alternative specifications. The greatest differences were seen in a scenario that used an alternative data source[17] for estimating tuberculosis-attributable mortality among tuberculosis survivors and a scenario that assumed that post-TB disability weights for RR-TB were the same as for RS-TB. For both scenarios, total RR-TB DALYs were estimated to be 15–16% lower than in the main analysis.

### Discussion
We assessed the global disease burden due to incident RR-TB in 2020, including mortality and morbidity during the disease episode as well as

**Table 3 | Global DALYs resulting from incident RR-TB 2020 in 2020, stratified by age group, sex, and HIV status**

| Stratum | RR-TB DALYs per 100,000[a] | DALYs per RR-TB case | Total RR-TB DALYs (millions) | Percent of global RR-TB burden (%) |
|---|---|---|---|---|
| **Age group** | | | | |
| 0–4 years | 68.7 (48.8, 96.0) | 34.1 (28.0, 39.9) | 0.47 (0.33, 0.65) | 6.7 (5.0, 8.8) |
| 5–14 years | 26.3 (17.9, 38.0) | 21.7 (16.3, 27.0) | 0.35 (0.24, 0.51) | 5.0 (3.8, 6.6) |
| 15–24 years | 89.3 (64.1, 116.7) | 17.8 (12.8, 22.6) | 1.08 (0.78, 1.42) | 15.6 (13.9, 17.0) |
| 25–34 years | 111.4 (84.6, 140.8) | 18.0 (13.6, 22.1) | 1.33 (1.01, 1.68) | 19.2 (17.9, 20.2) |
| 35–44 years | 129.2 (102.6, 157.8) | 18.3 (14.7, 21.8) | 1.34 (1.06, 1.64) | 19.3 (18.4, 20.1) |
| 45–54 years | 119.2 (97.4, 142.7) | 17.1 (14.2, 19.8) | 1.09 (0.89, 1.31) | 15.8 (14.9, 16.6) |
| 55–64 years | 107.1 (89.5, 126.0) | 14.9 (12.6, 16.9) | 0.75 (0.63, 0.89) | 10.9 (10.1, 11.8) |
| 65+ years | 70.2 (59.4, 81.6) | 10.4 (9.0, 11.7) | 0.52 (0.44, 0.60) | 7.5 (6.7, 8.3) |
| **Sex** | | | | |
| Male | 109.2 (85.8, 135.4) | 17.1 (13.7, 20.4) | 4.29 (3.37, 5.32) | 61.9 (57.9, 65.8) |
| Female | 68.1 (52.7, 85.3) | 17.5 (13.9, 20.8) | 2.64 (2.04, 3.31) | 38.1 (34.2, 42.1) |
| **HIV status** | | | | |
| HIV uninfected | 75.0 (58.9, 92.3) | 16.6 (13.1, 19.9) | 5.83 (4.57, 7.18) | 84.1 (80.0, 87.2) |
| HIV infected | 2,927 (2,151, 3,882) | 21.7 (17.3, 25.7) | 1.10 (0.81, 1.46) | 15.9 (12.8, 20.0) |

*DALYs* disability-adjusted life years, *RR-TB* rifampicin-resistant tuberculosis.
[a]Denominator represents the total global population of each stratum. Values in parentheses represent 95% uncertainty intervals.

the lifetime impact of post-TB sequelae among those surviving the disease. We found that RR-TB occurring in 2020 was responsible for 6.9 (5.5, 8.5) million DALYs, or 5.4% of the total DALYs due to all TB in 2020. An average of 17.3 (13.8, 20.6) DALYs were associated with each individual developing RR-TB, with nearly half of the total DALYs estimated for RR-TB accruing among tuberculosis survivors. While the South-East Asia Region was estimated to have the greatest absolute burden of RR-TB, former Soviet Union countries and southern African countries had the highest burden of RR-TB per population.

For former Soviet Union countries, this high RR-TB burden results from historical approaches to tuberculosis treatment that produced high resistance levels[18], with a large fraction of current tuberculosis cases experiencing resistance to rifampicin and, in some cases, many other tuberculosis drugs[1]. While approaches to tuberculosis diagnosis and treatment in the region have been formulated to address this challenge[19], it still represents a major barrier to effective tuberculosis care, and the poor outcomes experienced by individuals with drug-resistant tuberculosis (as compared to drug-susceptible disease) are reflected in the high RR-TB burden estimates.

For southern African countries, the high RR-TB burden estimates are likely related to the dual burden of tuberculosis and HIV in these countries. The relationship with HIV is notable, with HIV-infected individuals estimated to experience 40 times greater DALYs from RR-TB compared to HIV-uninfected individuals, a consequence of both elevated tuberculosis incidence for individuals with HIV and rapid disease progression and death in the absence of prompt, effective treatment. For southern African countries, the close overlap of tuberculosis and HIV epidemics may be exacerbated by poverty and health system weaknesses that make it difficult to achieve successful treatment outcomes for individuals with both HIV and RR-TB.

This study relies on accurate epidemiological data on drug resistance patterns across countries and population groups. This highlights the utility of efforts that have been made to develop and extend systems for monitoring drug resistance trends across many countries[20]. While for some countries we needed to impute data gaps using other evidence, the major sources of uncertainty in this study derive from the challenge of extrapolating future health outcomes for individuals with RR-TB. This additional uncertainty is reflected in the wider uncertainty intervals reported for post-TB DALYs, as compared to DALYs estimated for the disease period. While much of the additional DALYs per RR-TB case (as compared to RS-TB cases) resulted from

higher mortality during the disease episode, post-TB DALYs were also estimated to be higher for RR-TB, reflecting more prevalent and more severe sequelae following RR-TB. A substantial body of evidence now describes the elevated mortality and health burden experienced by tuberculosis survivors, motivating efforts to establish clinical standards[21] and systems of care for this large cohort of individuals[22]. However, some part of this excess morbidity and mortality will relate to pre-existing health conditions and behaviors that co-occur with tuberculosis. Moreover, given the concentration of tuberculosis in poor and marginalized communities, the lower health outcomes for tuberculosis survivors will also partly reflect the impact of socio-economic determinants of health that would still be present in the absence of tuberculosis. In this study, we extended a published approach to separating the causal impact of tuberculosis from these other factors[12], which requires several assumptions. For this reason, additional empirical research on the causal contribution of tuberculosis and RR-TB to future health outcomes is needed.

This research has several additional limitations. First, there is additional uncertainty around RR-TB outcomes in pediatric populations due to greater uncertainties in incidence estimates (related to the fundamental difficulties in detecting tuberculosis in this population)[23]. Uncertainty in pediatric estimates is compounded by limited data on post-TB outcomes for pediatric tuberculosis survivors, though the data that exist suggest persistent disease sequelae[24]. Second, our analysis makes assumptions about healthcare interventions and access for RR-TB survivors that may not hold in the future. In particular, the development and scale-up of effective rehabilitation interventions for post-TB lung disease could improve outcomes for RR-TB survivors[21,25] and reduce the burden relative to the estimates in this study. Similarly, the development and deployment of better diagnostics[26] and treatments[27] for RR-TB may reduce the excess disease burden experienced by individuals with RR-TB relative to drug-susceptible disease. In particular, the adoption of new all-oral 6-month regimens for RR-TB could produce major improvements in RR-TB treatment outcomes in the future. An MDR/RR-TB regimen composed of bedaquiline, pretomanid, linezolid and moxifloxacin (BPaLM) was included in a 2022 update to WHO treatment guidelines[2], and many countries have subsequently moved to introduce this regimen. Other short-course regimens are under development[28]. Given the better end-of-treatment outcomes achieved by the BPaLM regimen compared to older long-course regimens[27] and the fact that cure is achieved more rapidly, it is

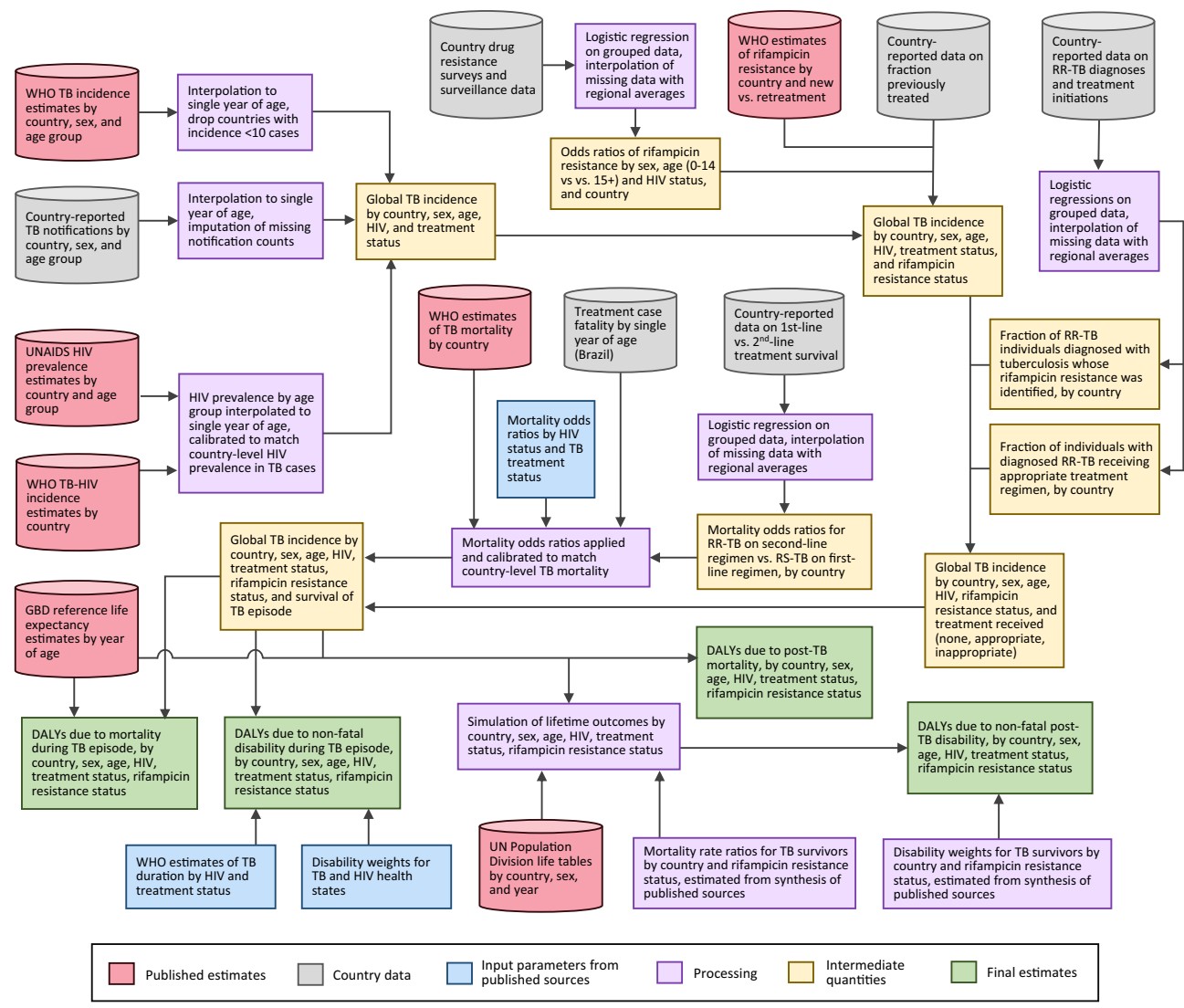

**Fig. 4 | Schematic of study inputs and estimation process.** DALYs disability-adjusted life years, RR-TB rifampicin-resistant tuberculosis, RS-TB rifampicin-susceptible tuberculosis.

possible that the introduction of new short-course RR-TB regimens will improve treatment outcomes and reduce post-treatment morbidity for RR-TB cohorts in the future.

This study suggests several directions for future research. First, empirical studies are needed that adopt quasi-experimental designs to separate the long-term effects of TB on post-TB health outcomes from the impact of comorbid conditions, health behaviors, and living conditions. These studies will be more valuable if they are able to describe how the causal effects of TB vary by individual-level factors such as *M. tuberculosis* strain, clinical presentation, and treatment regimen. A second priority is interventional studies that examine approaches to protect and restore lung function during TB disease treatment and identify best practices for providing long-term rehabilitation for TB survivors. Finally, policy analyses are needed to identify the most impactful and cost-effective approaches along the care continuum (prevention, diagnosis, treatment, and rehabilitation) for RR-TB disease.

RR-TB represents a major disease burden globally and a fundamental, lifelong health challenge for affected individuals. While the burden estimates reported by this study highlight the regions, countries, and population groups most impacted by RR-TB, they provide little guidance on the appropriate response to this burden. New interventions to facilitate the rapid detection and effective care for RR-

TB are urgently needed, as well as concerted efforts to expand access to the best current diagnostics and treatments and policy analyses that compare actions to prevent, treat, and rehabilitate RR-TB in order to devise the most impactful approaches for reducing RR-TB burden in the future.

## Methods
### Mathematical model
We adapted a previously published mathematical model of future health outcomes among the global cohort of individuals with tuberculosis disease[12]. This model stratifies the cohort developing tuberculosis in a single year by country, age, sex, and HIV status and tracks this cohort over the remaining lifetime of the affected individuals. Over this projection period, we recorded excess deaths and reductions in health-related quality of life causally attributable to TB or post-TB sequelae. For this analysis, we additionally stratified the model by the presence/absence of rifampicin resistance. For individuals without rifampicin resistance, we stratified the cohort according to whether the individual received or did not receive tuberculosis treatment. For individuals with rifampicin resistance, we stratified the cohort according to whether the individual (1) did not receive tuberculosis treatment, (2) received tuberculosis treatment with an inappropriate regimen (defined as the standard first-line regimen recommended for

rifampicin-susceptible tuberculosis), or (3) received treatment with an appropriate regimen (defined as a WHO-recommended RR-TB regimen). Analytic inputs and methods are described below and summarized in Fig. 4. Additional details are provided in Supplementary Methods.

## Data on individuals developing tuberculosis disease

Estimates of the total number of individuals developing tuberculosis in 2020 were extracted from epidemiological estimates produced by the WHO Global Tuberculosis Programme stratified by country, sex, and age group[29]. The fraction of cases receiving treatment was based on the number of notified tuberculosis cases within each country, sex, and age group for 2020, divided by the estimated incidence for each stratum. For countries with missing notification data, we used WHO-estimated tuberculosis treatment coverage. We removed countries with less than 10 estimated cases for 2020 (Supplementary Fig. S1), retaining 192 countries, representing 9.9 million tuberculosis cases (99.99% of all cases globally). We stratified this cohort by HIV status using WHO Global Tuberculosis Programme estimates of HIV prevalence among individuals with tuberculosis and additional epidemiological estimates from UNAIDS to decompose the TB-HIV population by age (details in Supplementary Information).

To stratify the cohort by rifampicin resistance status, we used country-level estimates of rifampicin resistance prevalence among new and previously treated cases and applied these estimates to country-reported data on the fraction of treated individuals with a history of prior tuberculosis. We used data from drug resistance surveys and routine surveillance activities to decompose the sex-, age-, and HIV-stratified cohort by the presence or absence of rifampicin resistance (details in Supplementary Information).

Within each stratum, we assumed that the fraction receiving any tuberculosis treatment was independent of rifampicin resistance. We estimated the fraction receiving a diagnosis of rifampicin resistance (among the population with rifampicin resistance diagnosed with tuberculosis) by dividing country-reported data on RR-TB diagnoses by WHO estimates of RR-TB incidence scaled by the fraction detected. We estimated the fraction of RR-TB initiated on second-line treatment by comparing data on RR-TB diagnoses and second-line treatment initiations for each country. Several countries had missing values for the data used to construct the study cohort. Supplementary Fig. S1 shows the availability of data for each country, and methods used to impute missing data are described in the Supplementary Information.

## Duration and disability of tuberculosis disease episode

We based assumptions around the duration of disability during the tuberculosis disease episode on values estimated by the WHO, stratified by treatment and HIV status (Supplementary Table S1). In the main analysis, we assumed the same duration for RS-TB and RR-TB and examined alternative assumptions in sensitivity analyses.

Disability weights for tuberculosis were based on current Global Burden of Disease Study estimates (Supplementary Table S1). For individuals with HIV and without tuberculosis, we averaged the disability weights for 'HIV: symptomatic, pre-AIDS' and 'HIV/AIDS: receiving antiretroviral treatment', weighted by the fraction of HIV-infected individuals receiving antiretroviral treatment in each country, as reported by UNAIDS for 2020. We calculated the incremental disability weight associated with tuberculosis as the difference between individuals with and without tuberculosis, stratified by HIV status.

## Fraction surviving the tuberculosis disease episode

We specified mortality odds ratios describing differences in survival probabilities by age, HIV, and tuberculosis treatment status. Mortality odds ratios by age were estimated from detailed data on case fatality among notified tuberculosis cases in Brazil (these data were used

because detailed data of this type are not widely available). Mortality odds ratios for HIV and for receipt vs. non-receipt of tuberculosis treatment were based on mortality risks estimated by the WHO Global Tuberculosis Programme. We allowed for differences in survival between individuals with RS-TB receiving a first-line regimen (assumed to be all RS-TB individuals), individuals with RR-TB receiving a second-line regimen, and individuals with RR-TB inappropriately receiving a first-line regimen, using country-reported data (details in Supplementary Information).

There is little evidence of survival among individuals with RR-TB inappropriately receiving a first-line regimen, and we calculated this value as a weighted average of survival probabilities for individuals with RR-TB receiving a second-line regimen and untreated individuals, with a wide prior distribution specified for this weighting parameter (Supplementary Table S1). We tested extreme values of 0.0 and 1.0 in sensitivity analysis. Individuals with RR-TB and RS-TB not receiving tuberculosis treatment were assumed to have the same mortality risks. Overall mortality risks were calibrated to reproduce country-specific case fatality rates reported by the WHO Global Tuberculosis Programme for 2020.

## Future mortality risks and disability for individuals surviving tuberculosis disease

Following Menzies et al.[12], we assumed that overall mortality differences between tuberculosis survivors and the general population could be decomposed into factors causally attributable to tuberculosis and pre-existing factors that would produce elevated mortality in the absence of tuberculosis and that should therefore be omitted from tuberculosis burden estimates. The product of these two effects (the overall mortality rate ratio for individuals with post-TB compared to the general population) was based on a systematic review of observational cohort studies (Supplementary Table S1)[8]. Estimates of the causal impact of tuberculosis on future mortality were estimated from excess COPD (chronic obstructive pulmonary disease) mortality among individuals surviving tuberculosis disease, following a published approach (details in Supplementary Information)[12]. These methods were also used to estimate post-TB disability weights that were applied to tuberculosis survivors. We assumed elevated mortality rates and disability weights among individuals surviving RR-TB vs. RS-TB based on studies reporting a higher prevalence and severity of tuberculosis sequelae among individuals surviving RR-TB as compared to RS-TB[13] (Supplementary Table S1). We conducted a sensitivity analysis re-estimating results with the conservative assumption that post-TB mortality rate ratios and disability weights for RR-TB are the same as for RS-TB.

## Outcomes

Health outcomes were quantified as the years of life lost due to premature mortality (YLLs) and the years of healthy life lost due to nonfatal disability (YLDs). For each modeled stratum, we calculated total tuberculosis-attributable DALYs as the sum of: (1) $YLD_{tb}$, representing DALYs lost to disability during tuberculosis disease episode; (2) $YLL_{tb}$, representing DALYs lost to mortality during the tuberculosis disease episode; (3) $YLD_{ptb}$, representing DALYs lost to post-TB disability among TB survivors; and (4) $YLL_{ptb}$, representing DALYs lost due to post-TB mortality.

## Uncertainty and sensitivity analysis

We specified probability distributions representing uncertainty in model parameters (Supplementary Table S1), and used second-order Monte Carlo simulation to generate 95% uncertainty intervals for study outcomes[30]. To do so, we re-estimated the model for each of the 1000 parameter sets created from a Latin hypercube sample from the parameter probability distributions and calculated intervals as the 2.5th and 97.5th percentiles of the resulting distribution. We also

calculated partial-rank correlation coefficients (PRCCs), describing the sensitivity of the estimate for total global DALYs due to RR-TB to uncertainty in individual parameters[31]. For parameters that varied at the country level (Supplementary Table S1), we calculated the equivalent global value and used these values when calculating PRCCs. Analyses were conducted in R (version 4.2.2).

We recalculated results for several alternative analytic assumptions: (1) the duration of RR-TB disease assumed to be 50% greater than for RS-TB, (2) the case fatality for RR-TB inappropriately treated with a first-line regimen assumed to be the same as for untreated individuals, (3) case fatality for RR-TB inappropriately treated with a first-line regimen assumed to be the same as for individuals treated with a second-line regimen, (4) case fatality for untreated RR-TB assumed to be greater than for RS-TB (OR = 1.5), (5) post-TB mortality rates based on estimates reported by Lee Rodriguez et al.[17], estimated from a retrospective cohort study that controlled for multiple demographic and clinical risk factors for mortality, and (6) post-TB mortality rates and disability weights for RR-TB assumed to be the same as for RS-TB. For each alternative specification, we estimated total global RR-TB DALYs, average DALYs per incident RR-TB case, and the fraction of global tuberculosis DALYs attributable to RR-TB.

### Reporting summary
Further information on research design is available in the Nature Portfolio Reporting Summary linked to this article.

## Data availability
Most data used in this study are publicly available. UN Population Division life tables are available here: https://population.un.org/wpp/. WHO TB data (country-reported data and estimated values) are available here: http://www.who.int/tb/country/data/download/en/. UNAIDS HIV estimates are available here: https://aidsinfo.unaids.org/. Global Burden of Disease Study disability weights are available here: http://ghdx.healthdata.org/record/ihme-data/gbd-2019-disability-weights. Brazil TB data are available here: https://datasus.saude.gov.br/transferencia-de-arquivos/. Copies of these datasets are available at https://doi.org/10.5281/zenodo.8339608. Historical data from routine drug resistance surveillance and surveys are not publicly available due to privacy issues and were shared by WHO for this analysis. Summary estimates of these data are published each year in WHO's Global TB Report (https://www.who.int/teams/global-tuberculosis-programme/tb-reports/global-tuberculosis-report-2022). Data can be provided on request to A.S.D. at deanan@who.int, with an expected response time of 2 weeks.

## Code availability
Analytic code for this analysis is available at https://doi.org/10.5281/zenodo.8339608.

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

## Acknowledgements

N.A.M. and T.C. were supported by a grant from the National Institute of Allergy and Infectious Diseases (R01AI146555). P.J.D. was supported by a fellowship from the UK Medical Research Council (MR/W029227/1). G.M.K. was supported by a fellowship from the UK Medical Research Council (MR/W026643/1). L.P.J. was supported by a grant from the National Institute of Allergy and Infectious Diseases (AI007433). R.M.G.J.H. received funding from the European Research Council (Action number 757699). Study sponsors had no role in study design; data collection, analysis, and interpretation; report writing; or the decision to publish.

## Author contributions

N.A.M., S.G.S., L.N.N., F.M., and T.C. conceptualized the study. B.W.A., A.S.D., P.J.D., R.M.G.J.H., L.P.J., G.K., J.M., and A.R. contributed to the study design. N.A.M. implemented the analysis. B.W.A., A.S.D., P.J.D., R.M.G.J.H., L.P.J., G.K., J.M., L.N.N., A.R., S.G.S., F.M. and T.C. reviewed and interpreted results. N.A.M. and T.C. drafted the manuscript. B.W.A., A.S.D., P.J.D., R.M.G.J.H., L.P.J., G.K., J.M., L.N.N., A.R., S.G.S., and F.M. edited the manuscript.

## Competing interests

The authors declare no competing interests.

## Additional information

¹Department of Global Health and Population, Harvard T. H. Chan School of Public Health, Boston, USA. ²Center for Health Decision Science, Harvard T. H. Chan School of Public Health, Boston, USA. ³Division of Pulmonology, Department of Medicine, Stellenbosch University & Tygerberg Hospital, Cape Town, South Africa. ⁴Global Tuberculosis Programme, World Health Organization, Geneva, Switzerland. ⁵School of Health and Related Research, University of Sheffield, Sheffield, United Kingdom. ⁶TB Modelling Group, TB Centre, London School of Hygiene and Tropical Medicine, London, United Kingdom. ⁷Department of Infectious Disease Epidemiology, London School of Hygiene and Tropical Medicine, London, United Kingdom. ⁸Harvard Interfaculty Initiative in Health Policy, Harvard University, Cambridge, USA. ⁹AMR Centre, Department of Infectious Disease Epidemiology, EPH, London School of Hygiene and Tropical Medicine, London, United Kingdom. ¹⁰National Heart & Lung Institute, Imperial College London, London, United Kingdom. ¹¹Division of Infectious Diseases and Tropical Medicine, Medical Centre of the University of Munich (LMU), Munich, Germany. ¹²German Centre for Infection Research (DZIF), Partner Site Munich, Munich, Germany. ¹³Unit Global Health, Helmholtz Zentrum München, German Research Center for Environmental Health (HMGU), Neuherberg, Germany. ¹⁴Department of Epidemiology of Microbial Diseases, Yale School of Public Health, New Haven, CT, USA. ¹⁵These authors contributed equally: Brian W. Allwood, Anna S. Dean, Pete J. Dodd, Rein M. G. J. Houben, Lyndon P. James, Gwenan M. Knight, Jamilah Meghji, Linh N. Nguyen, Andrea Rachow, Samuel G. Schumacher. ✉e-mail: nmenzies@hsph.harvard.edu

