## [Peer Review File · Nature Communications]

Global burden of disease due to rifampicin-resistant tuberculosis: a mathematical modelling analysisREVIEWER COMMENTS

Reviewer #1 (Remarks to the Author):

This is an outstanding modeling study with clear methodology and data sources, rigorous analyses, broad exploration of uncertainty/sensitivity, and clear public health relevance. The technical aspects of the study appear to be exceptionally strong.

My suggestions are only in regards to the framing and contextualization of the findings. The authors are welcome to take or leave what they find useful:

(1) The discussion around uncertainties and potential confounders in estimating post-TB burden after recovery from RR-TB was particularly impressive and thought-provoking. I wonder if the authors might be willing to go as far as to recommend studies or other evidence generation activities to reduce this uncertainty.

(2) The discussion of new oral RR-TB regimens is relegated to a phrase in Introduction and Limitations, but seems pivotal for contextualizing this study. Could the authors elaborate on new RR-TB regimens, how they compare to the regimens used over the time of analysis, what the implications of regimen change may be, and the current state of guidelines and coverage?

(3) Is there a way to contextualize the total DALYs attributable to RR-TB, e.g., as a % of total TB DALYs? Readers may not have great mental benchmarks for what is a large number of DALYs.

(4) The "Interpretation" section of the Abstract could be improved by focuses on non-obvious findings. That nearly half of burden being among TB survivors is a new result, but the other information conveyed here seem obvious to most people in the TB field even without the use of a sophisticated model. This space could be repurposed for any of the many non-obvious, high-impact results in the paper.

Reviewer identity: Anna Bershteyn, PhD

Reviewer #2 (Remarks to the Author):

I carefully reviewed the manuscript "Global burden of disease due to rifampicin-resistant tuberculosis: a mathematical modelling analysis". Using a mathematical model the authors estimated the burden of disease, expressed as disability adjusted life years (DALYs) accumulated over the lifetime for individuals developing tuberculosis in 2020 for 192 countries in 2020. The analysis is stratified by age, sex, HIV, and rifampicin resistance. This is a highly relevant research endeavor that underlines the role of antimicrobial resistance, adequate treatment and the continued relevance of tuberculosis.

Overall, the manuscript adequately outlines the methodological approach. I appreciate the detail the authors provide, particularly on model specifications, uncertainty estimations and sensitivity analyses. My only criticism relates to information on missing data and data quality. This information needs to be added to evaluate model results in light of data availability and quality.

Regarding results, I believe that the manuscript could be improved if the authors focused on several additional aspects: It would be highly interesting to understand, what burden could be avoided if RR-TB would be treated adequately. Furthermore, in addition to DALYs it would be interesting to see the contribution of YLLs versus YLDs. It would be particularly interesting if authors could display this by country. I was also wondering how the burden attributable to rifampicin resistance correlates with socio-demographic indicators, such as GDP, socio-demographic index or the health quality access index.

Response to reviewer comments for NCOMMS-23-30816

Reviewer comments have been renumbered for clarity. In addition to the edits made in response to reviewer comments, we have revised the manuscript to follow the Editorial Policy Checklist and the Reporting Summary, and confirmed the paper is consistent with the Nature guidance on reporting on sex and gender in research studies.

Reviewer #1

1. This is an outstanding modeling study with clear methodology and data sources, rigorous analyses, broad exploration of uncertainty/sensitivity, and clear public health relevance. The technical aspects of the study appear to be exceptionally strong. My suggestions are only in regards to the framing and contextualization of the findings. The authors are welcome to take or leave what they find useful.

Response: Thank you for these kind remarks. We have edited the manuscript to strengthen the framing and contextualization as described below.

2. The discussion around uncertainties and potential confounders in estimating post-TB burden after recovery from RR-TB was particularly impressive and thought-provoking. I wonder if the authors might be willing to go as far as to recommend studies or other evidence generation activities to reduce this uncertainty.

Response: Thank you for this suggestion, we have edited the Discussion section to provide pointers to future research needs:

“This study suggests several directions for future research. Firstly, empirical studies are needed that adopt quasi-experimental designs to separate the long-term effects of TB on post-TB health outcomes from the impact of comorbid conditions, health behaviors, and living conditions. These studies will be more valuable if they are able to describe how these causal effects of TB vary by individual-level factors such as M. tuberculosis strain, clinical presentation, and treatment regimen. A second priority is interventional studies, that examine approaches to protect and restore lung function during TB disease treatment, and that identify best practices for providing long-term rehabilitation for TB survivors. Finally, policy analyses are needed to identify the most impactful and cost-effective approaches along the continuum (prevention, diagnosis, treatment, and rehabilitation) of RR-TB disease.” [Page 16-17]

3. The discussion of new oral RR-TB regimens is relegated to a phrase in Introduction and Limitations, but seems pivotal for contextualizing this study. Could the authors elaborate on new RR-TB regimens, how they compare to the regimens used over the time of analysis, what the implications of regimen change may be, and the current state of guidelines and coverage?

Response: We have added text providing additional discussion of these new regimens:

“An MDR/RR-TB regimen composed of bedaquiline, pretomanid, linezolid and moxifloxacin (BPaLM) was included in a 2022 update to WHO treatment guidelines,² and many countries

have subsequently moved to introduce this regimen. Other short-course regimens are under development [Dookie et al 2022]. Given the better end-of-treatment outcomes achieved by the BPaLM regimen compared to older long-course regimens,³⁰ and that fact that cure is achieved more rapidly, it is possible that introduction of new short-course RR-TB regimens will improve treatment outcomes and reduce post-treatment morbidity for RR-TB cohorts in the future.” [Page 16]

New citation: Dookie N, Ngema SL, Perumal R, Naicker N, Padayatchi N, Naidoo K. The Changing Paradigm of Drug-Resistant Tuberculosis Treatment: Successes, Pitfalls, and Future Perspectives. Clin Microbiol Rev. 2022 Dec 21;35(4):e0018019.

4. Is there a way to contextualize the total DALYs attributable to RR-TB, e.g., as a % of total TB DALYs? Readers may not have great mental benchmarks for what is a large number of DALYs.

Response: The contribution of RR-TB to overall TB DALYs (5.4%) is currently described in the results and discussion section.

5. The “Interpretation” section of the Abstract could be improved by focuses on non-obvious findings. That nearly half of burden being among TB survivors is a new result, but the other information conveyed here seem obvious to most people in the TB field even without the use of a sophisticated model. This space could be repurposed for any of the many non-obvious, high-impact results in the paper.

Response: Thank you for this suggestion. We have edited the Interpretation section to focus on less obvious implications of the findings.

“Interpretation: While RR-TB causes substantial short-term morbidity and mortality, nearly half of the overall disease burden of RR-TB accrues amongst tuberculosis survivors. The substantial long-term health impacts amongst those surviving RR-TB disease suggest the need for improved post-treatment care and further justify increased health expenditures to prevent RR-TB transmission.” [Page 3]

Reviewer #2

6. I carefully reviewed the manuscript “Global burden of disease due to rifampicin-resistant tuberculosis: a mathematical modelling analysis”. Using a mathematical model the authors estimated the burden of disease, expressed as disability adjusted life years (DALYs) accumulated over the lifetime for individuals developing tuberculosis in 2020 for 192 countries in 2020. The analysis is stratified by age, sex, HIV, and rifampicin resistance. This is a highly relevant research endeavor that underlines the role of antimicrobial resistance, adequate treatment and the continued relevance of tuberculosis.

Response: Thank you for this summary.

7. Overall, the manuscript adequately outlines the methodological approach. I appreciate the detail the authors provide, particularly on model specifications, uncertainty

estimations and sensitivity analyses. My only criticism relates to information on missing data and data quality. This information needs to be added to evaluate model results in light of data availability and quality.

Response: Thank you for this positive assessment. We have added a new figure (Figure S1) describing the availability of data for each country and analytic input, so that readers can judge the data for each country. This figure is reproduced below.

Figure S1: Data availability by country and analytic input.

*Countries with less than 10 estimated TB cases for 2020 were excluded from analysis.

8. Regarding results, I believe that the manuscript could be improved if the authors focused on several additional aspects: It would be highly interesting to understand, what burden could be avoided if RR-TB would be treated adequately.

Response: Thank you for this suggestion. We agree this is an important question to address however we feel it would require a number of additional assumptions (such as consideration of transmission effects) and falls outside of the scope of the current analysis. We have added text highlighting this as a future research priority:

“Finally, policy analyses are needed to identify the most impactful and cost-effective approaches along the continuum (prevention, diagnosis, treatment, and rehabilitation) of RR-TB disease.” [Page 17]

9. Furthermore, in addition to DALYs it would be interesting to see the contribution of YLLs versus YLDs. It would be particularly interesting if authors could display this by country. I was also wondering how the burden attributable to rifampicin resistance correlates with socio-demographic indicators, such as GDP, socio-demographic index or the health quality access index.

Response: For the overall decomposition of DALYs into YLLs and YLDs, these are given in Table 1, and summarised in the results section text. While we do not give the breakdown by YLL/YLDs, country-level estimates of total RR-TB DALYs, RR-TB DALYs per 100,000, and DALYs per RR-TB case are given in the supplement. We agree these additional analyses may be of interest to readers, but worry that readers may interpret these relationships causally, which would be inappropriate based on our current analytic approach. We have added a paragraph to the discussion which speaks to the need for new study designs to better measure causal effects of TB on long-term health outcomes

“Firstly, empirical studies are needed that adopt quasi-experimental designs to separate the long-term effects of TB on post-TB health outcomes from the impact of comorbid conditions, health behaviors, and living conditions. These studies will be more valuable if they are able to describe how these causal effects of TB vary by individual-level factors such as M. tuberculosis strain, clinical presentation, and treatment regimen” [Page 16-17]